# Understanding the Role of a Cr Transition Layer in the Hot-Salt Corrosion Behavior of an AlSi Alloy Coating

**Tianxin Liu** [1,†], **Wei Chen** [1,†], **Suying Hu** [1,*], **Lin Xiang** [2], **Xu Gao** [1] **and Zhiwen Xie** [1,*]

1 School of Mechanical Engineering and Automation, University of Science and Technology, Anshan 114051, China
2 Southwest Technology and Engineering Research Institute, Chongqing 400039, China
* Correspondence: husuying@ustl.edu.cn (S.H.); xzwustl@126.com (Z.X.)
† The authors contributed equally to this work.

**Abstract:** The effect of a chromium (Cr) transition layer on the hot-salt corrosion behavior of an AlSi alloy coating was studied. Hot-salt corrosion experiments were performed at 650 °C and corrosion kinetic curves were plotted. The weight gain of the AlSi-coated samples increased to 0.89 mg/cm$^2$ at 100 h and then decreased steadily to 0.77 mg/cm$^2$ at 200 h. The weight of the AlSi-coated samples with the addition of a Cr transition layer increased immediately to 0.79 mg/cm$^2$ at 20 h and then gradually increased to 0.85 mg/cm$^2$ at 200 h. This Cr diffusion promoted the preferential creation of an Al$_2$O$_3$ layer, which effectively hindered the upward diffusion of Fe and also resulted in the production of a Cr$_2$O$_3$-SiO$_2$ layer, which impeded the multi-scale salt mixture's penetration. The Cr diffusion also caused a notable seal-healing effect, which healed the micro-pores. These oxidation and degradation reactions were considerably repressed by the high barrier properties of these oxide layers and the dense surface, resulting in the increased hot-salt corrosion resistance of the AlSi alloy coating. The current findings provide a feasible strategy for the design of a diffusion barrier layer of a thermal protective coating on martensitic stainless steel.

**Keywords:** stainless steel; alloy coating; hot-salt corrosion; diffusion; self-healing effect

## 1. Introduction

Martensitic stainless steel (MSS), a kind of strong and ductile material, is widely used in parts, such as exhaust passages around aeroengines and rocket fuel storage tanks, because of its high strength, good corrosion resistance, and oxidation resistance. [1–4]. However, due to the increasing demands of the energy industry in recent years, aeroengines could be eroded by oxidation and corroded by hot-salt mixtures when used in marine atmospheric environments, resulting in a significant reduction in the thermal stability and corrosion resistance of MSS. [5–7]. Long-term corrosion in a hot-salt environment severely limits their employment in broadening applications and their service life [8]. As a result, it is critical to devise an effective technique to improve the high-temperature corrosion resistance of MSS hot-end components [9].

Zhang et al. found that the corrosion resistance of stainless steel was significantly improved after a selective laser melting treatment with amorphous alloy additives [10]. Li et al. used plasma-nitriding technology to form a nitriding layer on the surface of stainless steel, which not only improved the surface hardness but also protected the metal from corrosion. [11]. Lei et al. used multi-arc ion plating technology to deposit a ZrN/ZrO$_2$ multilayer coating on the surface of stainless steel, which improved the corrosion resistance of the material [12]. Functional coatings are commonly used to improve the corrosion resistance of MSS hot-end components [13–23]. The aluminide coating demonstrated remarkable corrosion resistance in molten carbonate at 650 °C, which is attributed to the creation of a protective Al$_2$O$_3$ layer [13]. Furthermore, researchers have discovered that incorporating Co, Cr, and Si components into aluminide coatings can successfully

postpone corrosion deterioration [14–16]. Vu et al., for example, studied the corrosion behavior of an AlSi covering in sulfate and chloride solutions [14]. They discovered that the production of $SiO_2$ was the primary reason for the improved heat and corrosion resistance of MSS materials. The $SiO_2$ not only worked as a diffusion barrier for the S and Cl, but it also lowered the oxygen partial pressure at the coating–alloy contact. Furthermore, the Si doping accelerated the selective oxidation of Al to generate an $Al_2O_3$ protective layer. Unfortunately, during high-temperature corrosion exposure, the AlSi coatings suffered from significant elemental inter-diffusion [24–30]. According to the literature, the upward diffusion of Fe caused the creation of the ($Fe_2Al_5$ + FeAl) alloy phase, which resulted in significant volume shrinkage and coating cracking. These micro-cracks served as the diffusion pathways for the combined salt, speeding up the interaction between the developing $Cl_2$ and Al, resulting in significant coating deterioration and mass loss [28,29].

Up until now, these AlSi alloy coatings demonstrated a high potential for inhibiting the corrosion degradation of the MSS hot-end components when exposed to a high-temperature salt mixture; however, the significant upward diffusion of Fe severely damaged their internal structure and accelerated the degradation reaction. It is urgent to design a way to prevent elemental inter-diffusion between the coating and MSS substrate. This work constructed a thin Cr transition layer between the AlSi coating and the MSS substrate. The hot-salt corrosion behaviors of these AlSi coatings with and without a Cr transition layer were examined through extensive structural and test characterizations. The purpose of this work was to clarify the effects of the Cr layer on the corrosion degradation and diffusion behaviors of the AlSi coating. The barrier mechanism of the Cr layer in the corrosion behavior of the AlSi coating was also discussed.

## 2. Experimental

The substrate material was 1Cr13 martensitic stainless steel measuring 15 mm in length, 15 mm in width, and 5 mm in thickness. All samples were ground using sandpaper to remove surface impurities and then sandblasted with corundum particles with a mesh size of 240. These specimens were ultrasonically cleaned in an ethanol solution for twenty minutes. Arc ion plating was used to produce AlSi alloy coatings with and without a Cr transition layer on these MSS substrates (AIP). For this procedure, the arc targets were a high-purity AlSi alloy (80 wt.% Al and 20 wt.% Si) and Cr (99.9 wt.%), respectively. The working gas consisted of 99.9% pure argon. Before the coating deposition, all samples were cleaned by an arc-enhanced glow discharge to enhance the adhesion strength between the Cr and substrate. The parameters were 0.3 Pa argon pressure, 100 A current, −160 V bias voltage, and a time of 30 min. Initially, a Cr layer was formed on the substrate in an argon environment utilizing an arc current of 100 A, a pulse voltage of −100 V with a duty cycle of 60%, and a deposition duration of 20 min. The AlSi alloy layer was formed in an argon environment at a pressure of 1.0 Pa, an arc current of 80 A, a pulse voltage of −100 V with a 60% duty cycle, and a deposition duration of 75 min. AlSi coating without a Cr layer was produced for comparison using the same conditions described previously. The specifics of this coating procedure are detailed in Table 1.

**Table 1.** Preparation details of all coatings.

| Samples | Coatings |
| --- | --- |
| M1 | Single AlSi coating |
| M2 | Cr transition layer + AlSi top layer |

A high-temperature muffle furnace was used to examine the hot-salt corrosion behavior of all coatings. First, a 75 wt.% $Na_2SO_4$ + 25 wt.% NaCl supersaturated salt solution was uniformly applied to the surface of the preheated coated samples. To ensure that the salt film covered the surface of the coating uniformly, the salt application process was

repeated multiple times until the salt concentration of the sample reached 2–3 mg/cm$^2$. Subsequently, the coated samples, which had been coated with the mixed salt solution, were placed in a crucible and heated in a muffle furnace at 650 °C for 20 h. The etched samples were removed from the furnace and placed in air to cool to room temperature, the beaker was filled with deionized water heated to boiling in a water bath, and the samples were placed in the beaker and washed for 30 min. The salt test had a temperature, residence time, and heating rate of 650 °C, 200 h, and 10 °C/min. Throughout the 200 h hot-salt corrosion test, these samples were collected and weighed every 20 h, and the final results were used to plot the corrosion kinetic curves. The mass change of all as-deposited samples during the hot-salt corrosion process was measured with a precision of 0.0001 g using an electronic balance. For weighing purposes, samples must be washed in boiling deionized water until the water is nearly devoid of impurities. In order to reduce the experimental mass variation error, M1 and M2 samples were weighed using three sets of parallel specimens and the average mass was used as the final result. The crystal structures of all as-deposited and corroded coatings were identified using an X-ray diffractometer (XRD, X' Pert Powder, PANalytical B.V., Almelo, The Netherlands). Scanning electron microscopy (SEM, Zeiss ∑IGMA HD, Carl Zeiss, Jena, Germany) was used to examine the surface and cross-sectional morphologies of all as-deposited and corroded coatings. Energy dispersive spectroscopy (EDS) was used to characterize the elemental distributions of the deposited and corroded coatings using the SEM.

### 3. Results and Discussion

The XRD pattern of sample M2 is shown in Figure 1a. According to the earlier literature, these apparent diffraction peaks are mostly attributable to these Al and Si phases [31]. However, the Cr transition layer is barely visible in this XRD pattern, most likely because of the thick AlSi top layer, which prevents X-ray penetration. As illustrated in Figure 1b, sample M2 exhibits a rough surface morphology with some visible droplets and numerous white particles distributed on the coating's surface. The white particles are embedded in the gray matrix, as shown in the magnified view in Figure 1b. This coating contains a trace quantity of oxygen, which is primarily due to the leftover oxygen in the vacuum chamber. The Al, Si, and O contents in the B1 spot are 38.84 at.%, 58.31 at.%, and 2.85 at.%, showing a considerable Si enrichment. In contrast, the Al, Si, and O contents in the B2 spot are 71.65 at.%, 27.22 at.%, and 0.80 at.%, respectively, corresponding to an Al matrix. The cross-sectional SEM image clearly shows a two-layered structure, as illustrated in Figure 1c. The Cr and AlSi layers have thicknesses of around 3 and 25 m, respectively. According to the elemental mapping in Figure 1d, a continuous Cr transition layer distributes uniformly between the AlSi layer and the substrate.

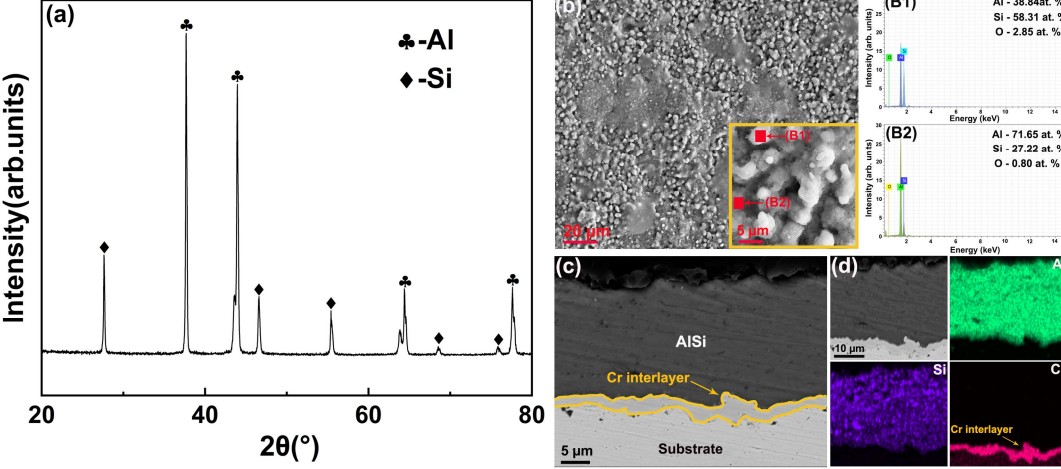

**Figure 1.** XRD pattern (**a**), surface SEM image (**b**), corresponding EDS spectrum (**B1**,**B2**), cross-sectional SEM image (**c**), and elemental mappings (**d**) of sample M2.

Figure 2a depicts the corrosion kinetic curves of all as-deposited samples during the hot-salt corrosion experiments. Sample M1 gains weight quickly at first but loses weight significantly afterward. This sample's weight increases to 0.89 mg/cm$^2$ at 100 h and then steadily declines to 0.77 mg/cm$^2$ at 200 h. In comparison, sample M2 has a strong weight rise tendency early on but a relatively moderate weight gain trend later on. The weight gain value of this sample immediately climbs to 0.79 mg/cm$^2$ at 20 h and gradually increases to 0.85 mg/cm$^2$ at 200 h.

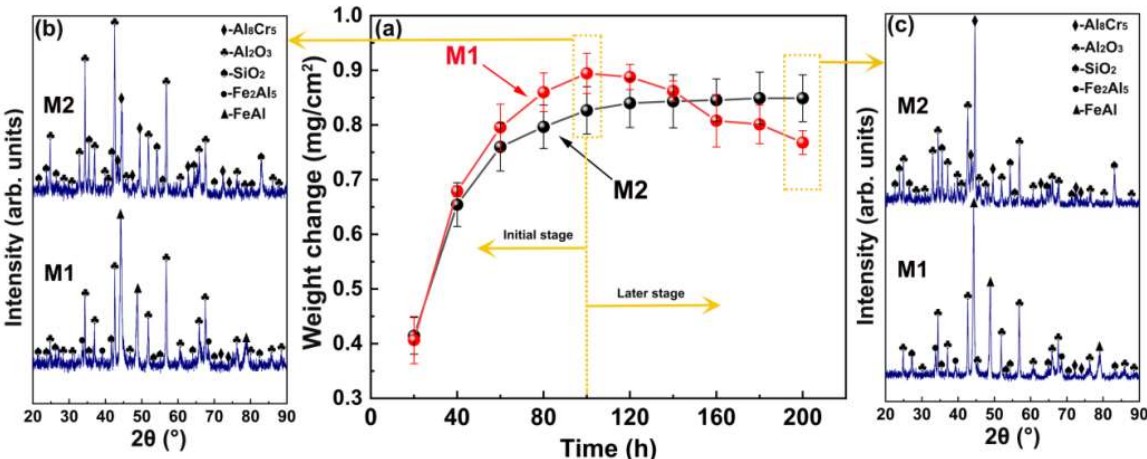

**Figure 2.** Corrosion kinetic curves (**a**), XRD patterns at 100 h (**b**), and XRD patterns at 200 h (**c**) of all as-deposited samples during the hot-salt corrosion tests.

The XRD patterns of all samples after 100 h of hot-salt corrosion testing are shown in Figure 2b. In sample M1, these $Al_8Cr_5$, $Al_2O_3$, and $SiO_2$ phases are detected, although a considerable number of $Fe_2Al_5$ and FeAl phases also occur, indicating a strong upward diffusion of Fe. In contrast, the phase compositions of sample M2 include $Al_8Cr_5$, $Al_2O_3$, and $SiO_2$, with no $Fe_2Al_5$ or FeAl, showing a good capacity to block Fe's upward diffusion. The presence of $Al_8Cr_5$ suggests a significant upward diffusion of Cr. As illustrated in Figure 2c, a comparable composition of $Al_8Cr_5$, $Al_2O_3$, $SiO_2$, $Fe_2Al_5$, and FeAl phases was discovered in sample M1 after a 200 h hot-salt corrosion test, but the relative intensities of these FeAl peaks are clearly elevated, indicating a worsened upward diffusion of Fe. After a 200 h hot-salt corrosion test, Sample M2 is mainly composed of $Al_8Cr_5$, $Al_2O_3$, and $SiO_2$. However, the intensities of the $Al_8Cr_5$ peaks at 200 h are higher than at 100 h, signifying a growing upward diffusion of Cr.

The surface SEM pictures and matching EDS spectra of all as-deposited coatings after a 100-h hot-salt corrosion test are shown in Figure 3. Sample M1 has a loose surface morphology, as seen in Figure 3a,b, with numerous pores and fissures on its surface. The EDS spectrum in Figure 3c shows that the corroded surface is composed of 32.67 at.% Al, 42.95 at.% Si, 1.08 at. Cr, 9.78 at.% O, and 13.52 at.% Fe. The elevated Fe concentration confirms the XRD characterization of a considerable Fe upward diffusion. Sample M2 has a loosely granular surface with multiple particles, as seen in Figure 3d,e. There are a lot of apparent micro-pores between these randomly scattered fragments. According to the EDS spectrum in Figure 3f, the corroded piece contains 30.91 at.% Al, 0.83 at.% Si, 2.36 at.% Cr, 65.79 at.% O, and 0.11 at.% Fe. Notably, a minor upward diffusion of Cr occurs during the hot-salt corrosion test, whereas the tiny Fe concentration suggests that this element has little upward diffusion during the hot-salt corrosion test.

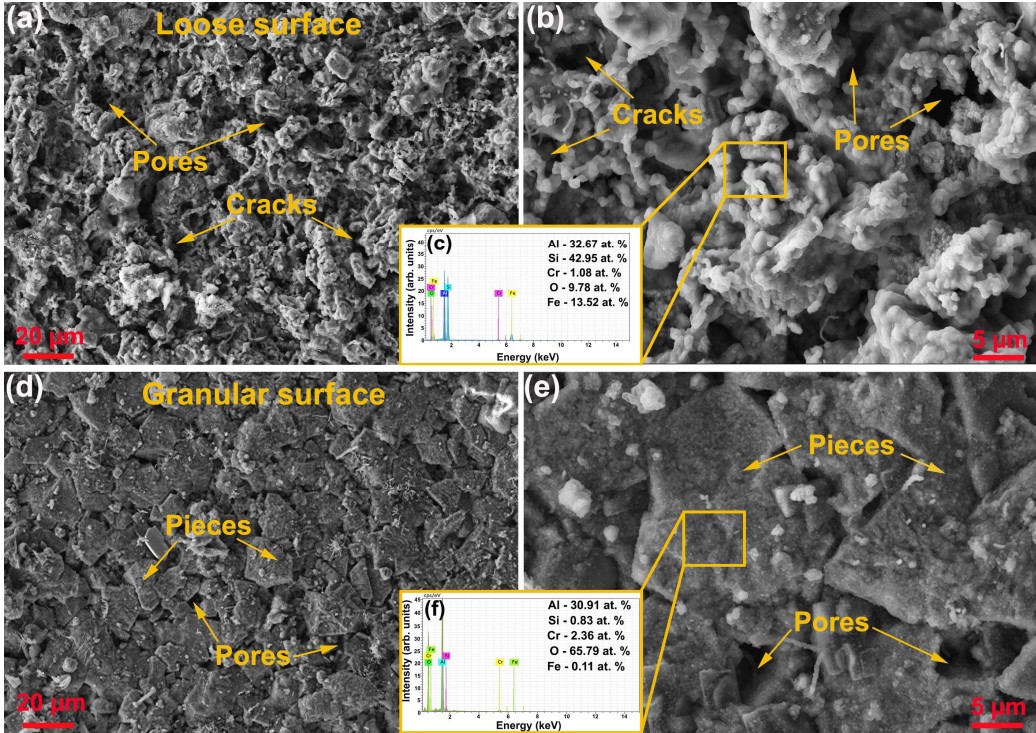

**Figure 3.** Surface SEM images with the corresponding EDS spectra of all samples after 100 h hot-salt corrosion tests: (**a**–**c**) M1; (**d**–**f**) M2.

The cross-sectional SEM pictures and element mappings of all samples following a 100 h hot-salt corrosion test are shown in Figure 4. Sample M1 has a loose cross-sectional morphology, as shown in Figure 4a, and some noticeable large-scale unfilled pores are seen in this corroded coating, showing that substantial structure degradation occurs during the hot-salt corrosion test. Furthermore, a significant upward diffusion of Fe and Cr is observed in the element mappings of Figure 4b, which agrees with the XRD results shown in Figure 2b. Sample M2 has a very dense cross-sectional morphology, as illustrated in Figure 4c,d, but it also has a clear upward diffusion of Cr. In the coating, a continuous mixed layer of $Cr_2O_3$ and $SiO_2$ forms. Furthermore, a continuous $Al_2O_3$ layer forms between the $Cr_2O_3$-$SiO_2$ layer and the substrate, thus preventing the upward diffusion of the Fe element.

The surface SEM images and matching EDS spectra of all as-deposited samples following 200 h hot-salt corrosion testing are shown in Figure 5. Sample M1 has a looser surface morphology than sample 100 h, as shown in Figure 5a,b, and numerous large-scale pores develop on the corroded surface. The EDS spectrum in Figure 5d shows that the corroded texture is composed of 48.07 at.% Al, 5.11 at.% Si, 0.28 at.% Cr, 20.94 at.% O, and 25.60 at.% Fe. The increased Fe concentration indicates a faster upward diffusion at a later stage. As demonstrated in Figure 5d,e, sample M2 has a denser surface morphology following a 200 h hot-salt corrosion test, with plenty of gray particles appearing on the coating's surface and further sealing these micro-cracks and pores. The gray particle's composition is 0.71 at.% Al, 0 at.% Si, 32.97 at.% Cr, 65.97 at.% O, and 0.34 at.% Fe, as seen in the EDS spectrum in Figure 5f. Notably, an accelerated upward diffusion of Cr occurs, which leads to a major healing impact later on.

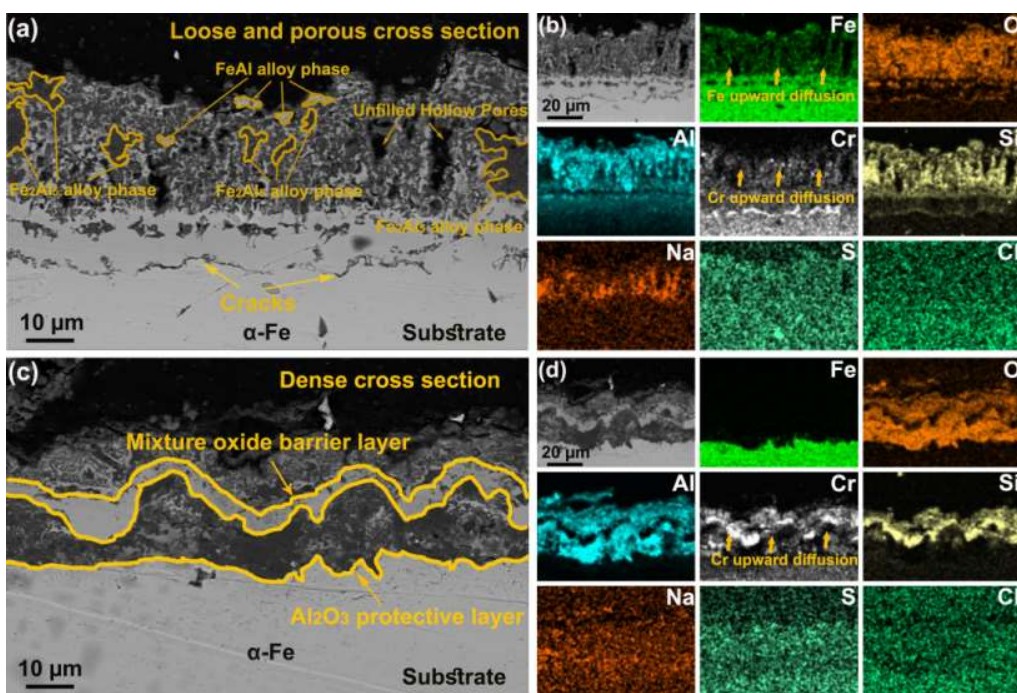

**Figure 4.** Cross-sectional SEM images and corresponding element mappings of all as-deposited coatings after 100 h salt corrosion test: (**a**,**b**) M1; (**c**,**d**) M2.

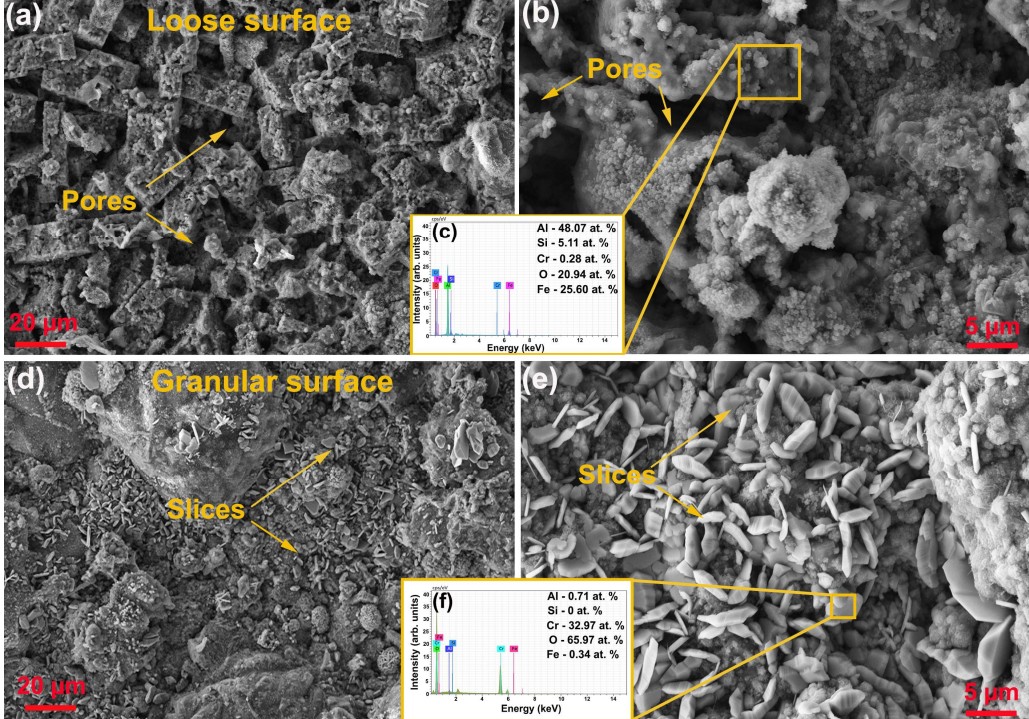

**Figure 5.** Surface SEM images with the corresponding EDS spectra of all samples after 200 h hot-salt corrosion tests: (**a**–**c**) M1; (**d**–**f**) M2.

Figure 6 shows the cross-sectional SEM images and element mappings of all as-deposited coatings after a 200 h salt corrosion test. As demonstrated in Figure 6a,b, sample M1 suffers from exacerbated corrosion damage later in the process, as evidenced by the more severe upward diffusion and enrichment of the Fe atoms. A loose surface layer is detected on the top of the corroded coating, which corresponds to the porous surface

depicted in Figure 6a. However, these pores are barely visible in the texture under the loose top layer, but significant areas of dendritic Al$_2$O$_3$ are distributed in this texture, which is most likely the result of the degradation process caused by salt mixture permeation.

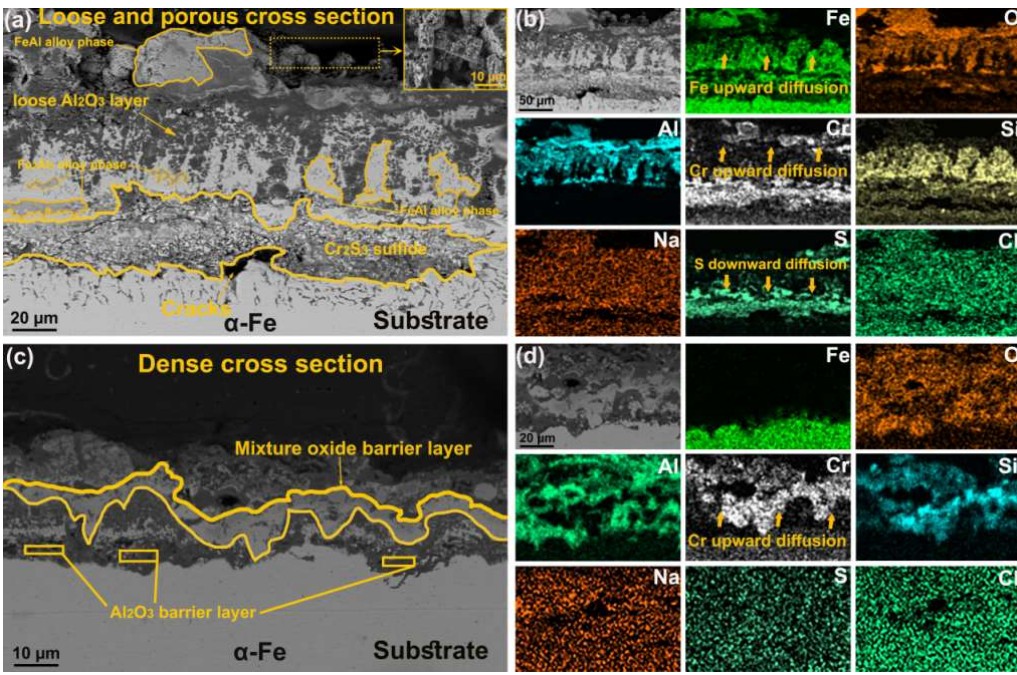

**Figure 6.** Cross-sectional SEM images and corresponding element mappings of all as-deposited coatings after 200 h salt corrosion test: (**a**,**b**) M1; (**c**,**d**) M2.

Furthermore, apparent enrichment zones of the S atoms appear in the substrate matrix, indicating that considerable internal vulcanization occurs later in the process. According to the XRD results in Figure 2 and the previous literature [20,22,30], the upward diffusion of the Fe atoms causes the formation of the Fe$_2$Al$_5$ and FeAl alloy phases in the coating [20,22], whereas the downward diffusion of the S atoms causes the formation of the Cr$_2$S$_3$ sulfide in the substrate matrix [28]. Sample M2 has a reasonably dense cross-sectional morphology, with no micro-cracks or internal vulcanization, as seen in Figure 6c,d, showing better hot-salt corrosion resistance. At a later stage, the upward diffusion of Fe is rare, but the upward diffusion of Cr is well-defined and results in the production of a mixed (Cr$_2$O$_3$ + SiO$_2$) layer. Furthermore, a continuous Al$_2$O$_3$ layer forms between the mixed oxide layer and the substrate forming a significant barrier to the upward passage of Fe.

Clearly, the Cr transition layer significantly impacts the high-temperature hot-salt corrosion resistance of AlSi alloy coatings. Figure 7 depicts the probable hypotheses for the corrosion damage mechanisms of various coatings. Sample M1 has poor hot-salt corrosion resistance, most likely due to the severe Fe upward diffusion and multi-scale salt mixture downward diffusion. Sample M1 undergoes oxidation, as illustrated in Figure 7a, resulting in the creation of a dense Al$_2$O$_3$-SiO$_2$ layer and a weight gain. Meanwhile, the molten salt mixture spreads throughout the coating. According to Equations (1) and (2) [32], a eutectic reaction occurs between Na$_2$SO$_4$ and NaCl. According to Equation (3), Cl$_2$ combines with Al to generate gaseous AlCl$_3$ [33,34]. Following that, according to Equation (4) [35], a component of AlCl$_3$ combines with O$_2$ to generate Al$_2$O$_3$, but another fraction of AlCl$_3$ inevitably overflows, resulting in weight loss and a porous surface (see Figure 7a (A1)).

$$Na_2SO_4 = Na_2O + SO_2 + 1/2O_2 \tag{1}$$

$$2NaCl + SO_2 + O_2 = Na_2SO_4 + 1/2Cl_2 \tag{2}$$

$$2Al + 3Cl_2 = 2AlCl_3 \tag{3}$$

$$4AlCl_3 + 3O_2 = 2Al_2O_3 + 6Cl_2 \tag{4}$$

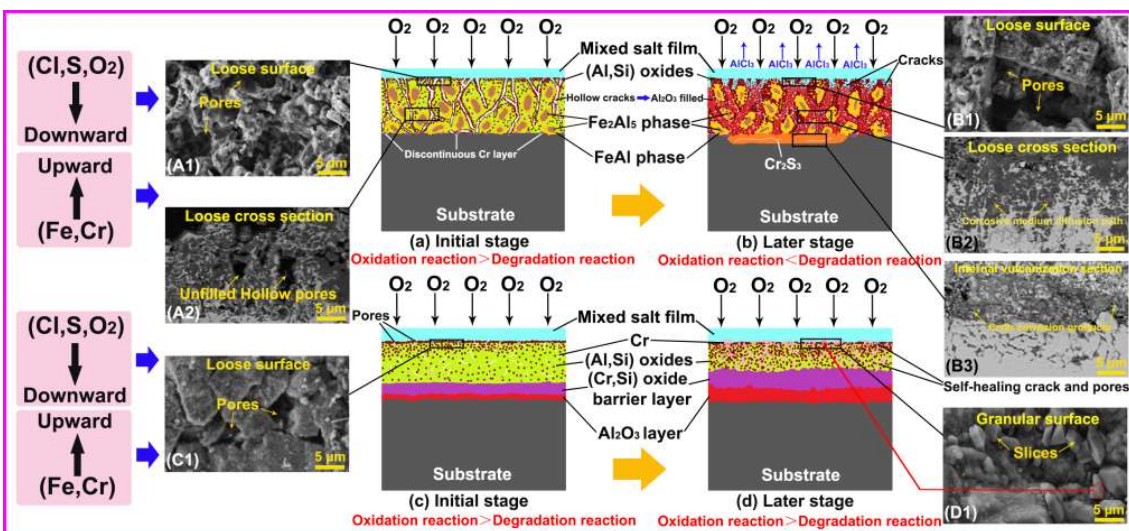

**Figure 7.** Schematic diagram of the degradation damage mechanisms of the as-deposited coatings during the hot-salt corrosion tests: (**a,b**) M1; (**c,d**) M2; SEM images of the surface of sample M1 after 100 h and 200 h hot salt corrosion: (**A1,B1**), SEM images of the cross section of sample M1 after 100 h and 200 h hot salt corrosion: (**A2,B2,B3**); SEM images of the surface of sample M2 after hot salt corrosion for 100 h and 200 h: (**C1,D1**).

Furthermore, Cr doping has been shown to cause the formation of a full $Al_2O_3$ layer in Fe–Al alloys [36]. However, the Cr concentration of the substrate matrix is insufficient, as illustrated in Figure 7a, where a continuous $Al_2O_3$ layer appears sporadically at the interface of the AlSi coating and substrate. As a result of the visible element diffusion indicated by the XRD and SEM data during the hot-salt corrosion test, these Fe diffuse into the coating and first react with Al to generate $Fe_2Al_5$ according to Equation (5) [22]. Furthermore, according to Equation (6), these adequate Fe react with previously created $Fe_2Al_5$ to form FeAl. These FeAl and $Fe_2Al_5$ alloy compositions have been shown to cause volume shrinkage, which results in many micro-cracks [37].

$$2Fe + 5Al = Fe_2Al_5 \tag{5}$$

$$3Fe + Fe_2Al_5 = 5FeAl \tag{6}$$

During the entire hot-salt corrosion test, the oxidation and degradation reactions occur simultaneously, but the upward diffusion rate of Fe is relatively slow at the beginning, as shown in Figure 7(A2). Sample M1 suffers from very slight salt mixture penetration and numerous unfilled hollow pores appear in this coating. As a result, the oxidation process dominates the degradation reaction governed by salt mixture penetration, and a continuous weight growth trend is shown in sample M1's first kinetic curve.

However, as illustrated in Figure 7b, an intensified upward diffusion of Fe occurs later in the process, resulting in a substantial volume fraction of FeAl and $Fe_2Al_5$ alloy compounds forming in this coating, causing severe volume shrinkage and coating cracking. These micro-cracks serve as the diffusion routes, allowing the salt combination to penetrate more quickly. As a result, as shown in Figure 7(B2,B3), the multi-scale penetration of the salt combination produces severe coating degradation and substrate vulcanization, resulting in the creation of the $AlCl_3$ and $Cr_2S_3$ products. The gaseous $AlCl_3$ diffuses outward along these pores and reacts with $O_2$ to form loose $Al_2O_3$, as evidenced by the $Al_2O_3$ branch-like morphology (Figure 7(B2)), but the overflow of the $AlCl_3$ increases significantly with the aggravated degradation reaction, eventually leading to a continuous weight loss and porous surface morphology (Figure 7(B1)) of sample M1 at a later stage.

Sample M2, on the other hand, exhibits increased hot-salt corrosion resistance, which could be attributed to the creation of a composite diffusion barrier layer and a dense surface. As shown by the SEM data in Figure 6c,d, there is a significant upward diffusion of Cr, which leads to creating a continuous $Cr_2O_3SiO_2$ mixture layer. The Cr layer also promotes the formation of a dense $Al_2O_3$ layer between the $Cr_2O_3$–$SiO_2$ layer and the substrate [28,36]. Sample M2 suffers from simultaneous oxidation and degradation reactions during the initial and subsequent hot-salt tests, as shown in Figure 7c,d, but this interface $Al_2O_3$ barrier layer effectively prevents the upward diffusion of Fe, suppressing the formation of the ($FeAl + Fe_2Al_5$) alloy compounds and severe volume shrinkage thus, structure fractures hardly appear in this coating.

Furthermore, this dense $Cr_2O_3$–$SiO_2$ mixture layer provides a significant barrier capability by preventing the salt mixture's downward diffusion. Numerous micro-cracks and pores emerge on the surface of the coating (Figure 7(C1)), which advantages a substantial oxidation process, resulting in a quick weight rise trend at the beginning. However, as demonstrated in Figure 7(D1), the upward diffusion of Cr is clearly accelerated with increasing test duration, resulting in the preferential formation of a dense $Al_2O_3$ layer [36] and a substantial self-healing effect. As a result, the strong synergistic effect of these continuously growing oxide layers and the dense surface provides an outstanding barrier capability by preventing multi-scale salt mixture penetration and the downward diffusion of $O_2$, resulting in a significant decrease in oxidation and degradation reactions. As a result, the accelerated kinetic curve of sample M2 shows an extremely sluggish weight gain trend.

## 4. Conclusions

This study examined the effect of the Cr transition layer on the hot-salt corrosion behavior of AlSi alloy coatings. This coating without a Cr layer experienced substantial volume shrinkage and coating breaking due to the strong upward Fe migration. These micro-cracks on a vast scale enhanced the multi-scale penetration of the salt mixture, resulting in accelerated corrosion deterioration and weight loss. Due to the upward diffusion of Cr, the addition of a Cr transition layer caused the preferred growth of an $Al_2O_3$ layer, the production of a $Cr_2O_3$-$SiO_2$ mixture layer, and a pronounced seal-healing effect on the dense surface. The $Al_2O_3$ layer successfully restricted the upward diffusion of Fe, but the $Cr_2O_3$-$SiO_2$ mixture layer significantly stopped the salt combination from penetrating downward. The synergistic effect of these oxide layers and the dense surface layer functioned as a strong barrier, inhibiting the oxidation and degradation reactions, resulting in the enhanced AlSi alloy coating's resistance to hot-salt corrosion.

**Author Contributions:** Conceptualization, S.H. and Z.X.; methodology, W.C.; investigation, T.L. and X.G.; and supervision, L.X. All authors have read and agreed to the published version of the manuscript.

**Funding:** This work was funded by the University of Science and Technology Liaoning Talent Project Grants (601011507-07), the Southwest Institute of Technology and Engineering Cooperation fund (HDHDW5902020103), and the Graduate Education Reform and Science and Technology Innovation Project of the University of Science and Technology Liaoning (LKDYC202113).

**Institutional Review Board Statement:** Not applicable.

**Informed Consent Statement:** Not applicable.

**Data Availability Statement:** No new data were created or analyzed in this study. Data sharing is not applicable to this article.

**Conflicts of Interest:** The authors declare no conflict of interest.

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
