# Peer review of "Understanding the Role of a Cr Transition Layer in the Hot-Salt Corrosion Behavior of an AlSi Alloy Coating"

_coatings, doi:10.3390/coatings12081167_

Round 1

Reviewer 1 Report

The article “Understanding the role of Cr transition layer on the hot salt corrosion behavior of AlSi alloy coating” by Tianxin Liu, Wei Chen, Suying Hua, Lin Xiang, Xu Gao, Zhiwen Xie describes to method to enhance the stability of protective AlSi coating for martensitic stainless steel. The authors mention that AlSi was already described, but it was unstable in highly concentrated salt solutions (mimicking the marine environment). And the authors mention that the reason for the instability is the migration of Fe. As the authors understand the mechanism underlying coating degradation, they propose a method to enhance the stability of the coating by introducing the intermediate Cr layer.

The authors perform extensive characterization of the samples with and without the protective Cr layer including SEM, XRD and EDS. It is remarkable that the authors limit the number of characterization methods which they use. Instead, they extract maximum structural information from each method. The authors elaborate the protective mechanism of the coating as well which is another strong side of this paper.

In my opinion, this article can be published in Coatings journal after a minor revision.

To enhance the quality of the manuscript, the authors can mention several methods described in the literature that allow for higher stability of metals in salt solutions.

Reviewer 2 Report

1) Title : Ok

2) Abstract : Short add more results with numbers

3) Keywords : ok

4) for references on the text [1-4]. not . [1-4] fix this matter in all your manuscript

5) Table1 splitted between two pages 

6) Fellow references template and update them by adding 2022

Reviewer 3 Report

This study reported that the effect chromium transition layer on the hit salt corrosion behavior of the AlSi alloy. The presence of Cr improved the preferential creation of an Al2O3 layer. However, this study needs to resolve some issues before acceptance.

-Introduction should be more general, special first par. Thus, authors should add more information regarding anticorrosion such doi.org/10.1016/j.pmatsci.2020.100663, doi.org/10.1016/j.molliq.2022.119513 , and doi.org/10.1016/j.apmt.2021.101142.

Should clarify the novelty from this work.

Does the inner layer (Cr interlayer) has high porosity and how Cr improve the adhesion.  

Which morphology is the best one for anticorrosion?

If possible, please add electrochemical performance such as PDP and EIS.

Round 2

Reviewer 3 Report

Present form is acceptable in coating